nanotechnology/materials science

sulfur-free synthesis, $Cu_{3-x}Te_2$, spheroids and planar squares, nanocrystals

**Author for correspondence:**
Jiang Du
e-mail: 0210927@163.com

# Sulfur-free synthesis of size tunable rickardite ($Cu_{3-x}Te_2$) spheroids and planar squares

Guanwei Jia[1,2], Chengduo Wang[1], Peixu Yang[1], Jinhui Liu[1], Weidong Zhang[1], Rongbin Li[4], Shaojun Zhang[1] and Jiang Du[1,3]

[1]Henan Province Industrial Technology Research Institute of Resources and Materials, Zhengzhou University, Zhengzhou 450001, People's Republic of China
[2]School of Physics and Electronics, Henan University, Kaifeng 475004, People's Republic of China
[3]Department of Chemical Engineering, Texas Materials Institute, Center for Nano- and Molecular Science and Technology, The University of Texas at Austin, Austin, TX 78712, USA
[4]School of metallurgical and Ecological Engineering, University of Science and Technology Beijing, Beijing 100083, People's Republic of China

JD, 0000-0001-8949-6230

We report a novel synthesis of monodisperse samples of copper telluride with crystallinity and stoichiometry corresponding to forms of rickardite, $Cu_{3-x}Te_2$ ($x < 1$). This synthesis makes use of a ligand balanced reaction to allow control over shape and size by varying the relative and absolute concentration of oleylamine to stearic acid. The rickardite samples presented here display size dependent plasmon peaks in the near infrared and direct energy band gaps between 1.7 and 2.3 eV. As such they may find utility in photovoltaic, thermoelectric or as novel optical materials for study of surface plasmons.

## 1. Introduction

Recently, solar cells using semiconductor nanocrystal (NC) films have shown great promise for the production of low-cost, high efficiency photovoltaic (PVs) devices. Colloidal nanocrystals [1,2] and nanowires [3–5] can be well controlled and deposited by low temperature, non-vacuum methods such as ink-jet printing [6], spray coating [7], soft templates [8] and roll-to-roll printing [9], while allowing tuning of the size, band gap, conductivity and crystallinity of the films [10–20]. As such PVs made from semiconductor NCs have a distinct advantage over current, state-of-the-art PVs made using thin film production methods. To date, photovoltaic devices (PVs) have been made from colloidal nanocrystals of PbSe, PbS, CdSe, CuIn(S,Se)₂, Cu(InGa)Se₂, CZTS, CdTe and TiO₂ colloidal nanocrystals [7,21–24].

This article has been edited by the Royal Society of Chemistry, including the commissioning, peer review process and editorial aspects up to the point of acceptance.

**Figure 1.** Standard reaction results. (*a*) Histograph and transmission electron micrographs of copper telluride nanocrystals from the 'standard' reaction. (*b*) NCs appear to be crystalline oblate spheres. (*c*) X-ray diffraction patterns (XRD) are matched by computer to those of $Cu_{2.86}Te_2$, an orthorhombic form of rickardite. Energy dispersive secondary X-ray (EDX) measurements on a sample on holey-carbon TEM grids show the sample to be purely copper (*d*) and tellurium (*e*) (dark spots are holes in the grid).

Given the interest in using copper based materials in PV applications, there is an effort to identify new nanomaterials to develop 'greener' [25] nanocrystal-based PVs. Copper tellurides, like many cupric chalcogenides ($Cu_xE_y$, E = S, Se, Te), occur in several stoichiometric and crystalline forms, and tend to be p-type materials with broad absorbance across the visible spectrum [26–28]. Forms of copper telluride have been used to make thin films [29], nanowires [30] and nanocrystals [10,26–28,30] and have been investigated recently as model systems for studying plasmon physics in quantum dots [10,31]. Forms of aggregated nanocrystalline rickardite [32,33] were reported in the late 1990s but without size or shape control, while monodisperse weissite and vulcanite have recently been reported [10,26,31]. Here we report the synthesis of highly monodisperse rickardite ($Cu_{3-x}Te_2$, $x < 1$) nanocrystals with non-hexagonal symmetry, produced by arrested precipitation in a high boiling solvent. With a direct band gap tunable between 1.7 and 2.3 eV, these materials have the potential to be used in a variety of photovoltaic, optical switching or thermoelectric [34] applications. Furthermore, by controlling reaction conditions such as relative ligand concentrations, we can demonstrate control over size, dispersity and crystal structure.

## 2. Material and methods

In a typical synthesis, 1 mmol of copper acetylacetonate is placed in a 3-neck flask with 12 ml of octadecene and 3 mmol of stearic acid. The flask is degassed under vacuum at 100°C for 1 h, after

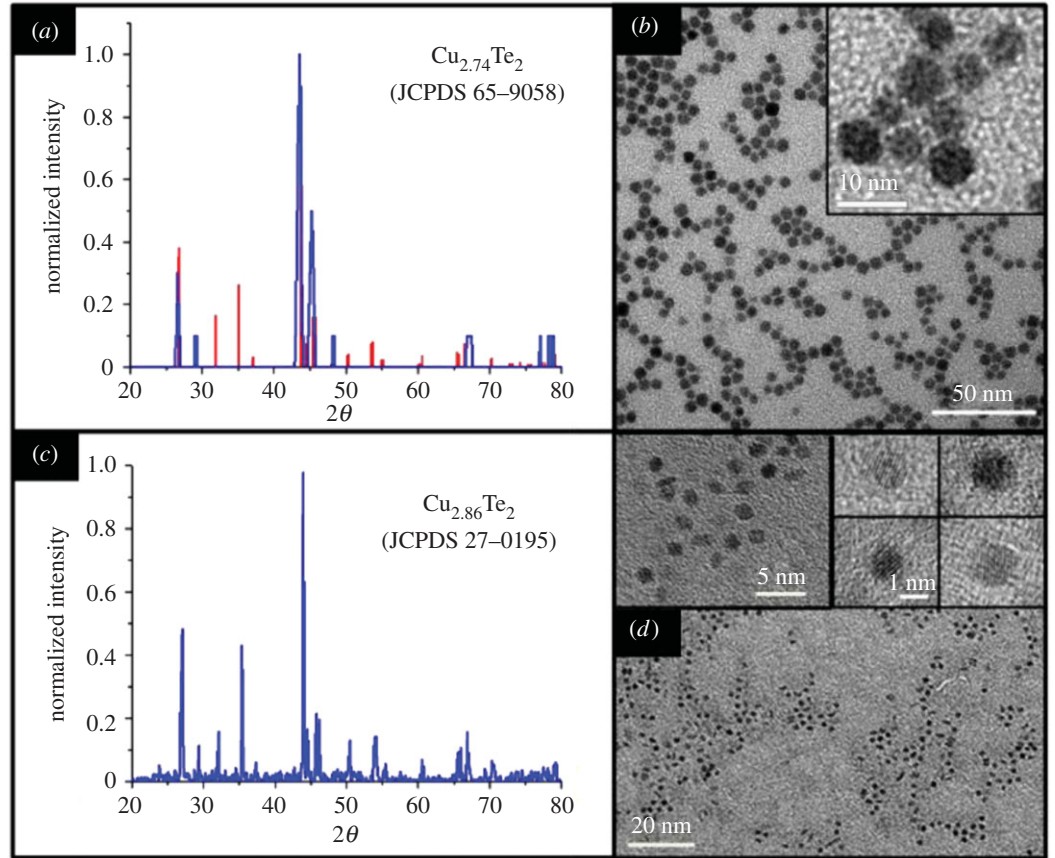

**Figure 2.** Size control. (*a*) XRDs of each match slightly different crystal structures, although both symmetries are within the same family and hence represent slightly more or less symmetric forms of the material. TEMs of rickardite NCs from the one-pot (*b*) and oleylamine syntheses (*c*) showing size control over the NCs. (*d*) The one-pot synthesis appears to be less crystalline and is difficult to match accurately.

which 0.5 ml of oleylamine is added, at which point the clear copper solution turns from blue-green to dark blue, indicating formation of a copper oleylamine complex. The solution is degassed at 100°C for a further 30 min after which it is placed under 1 atm of nitrogen and heated to the injection temperature, typically 165°C. 4 ml of a 0.25 M solution of trioctylphosphine telluride is then quickly injected into the solution to cause nucleation and the nanocrystals are grown for approximately 5 min at 180°C before the reaction is cooled quickly to room temperature. Nanocrystals can be purified up to three times using the standard solvent/non-solvent method, with ethanol as the initial non-solvent and toluene as the dispersant.

# 3. Results and discussion

The synthesis described above generated monodisperse, spheroidal nanocrystals with average diameter of $9.2 \pm 1.1$ nm (approx. 11%) variation in the major axis (figure 1*a*,*b*). Based on X-ray diffraction (XRD) measurements (figure 1*c*) these particles appear to have orthorhombic structure of rickardite, $Cu_{2.86}Te_2$, and are clearly neither weissite ($Cu_{2-x}Te$, JCPDS 10-0421) nor vulcanite (CuTe, JCPDS 65-4264) nor other observed forms of copper telluride, copper tellurium oxide or any combinations or subsets thereof. Elemental analysis (figure 1*d*,*e*) indicates that they are a copper deficient form of the crystal as the ratio of copper to tellurium was measured to be 2.45 : 2 ($x = 0.55$) via energy dispersive X-ray (EDX) spectrum measurements. UV-vis-IR absorption measurements (figure 4*e*) show strong absorption in the visible part of the spectrum dropping off rapidly near the band edge. The band gap for these particles was estimated to be around 2.2 eV using the linear region of the Tauc plot for a direct band gap semiconductor, which also reveals a significant Urbach tail indicating a large density of trap states near the band edge. At energies below the band gap, a broad plasmon absorption peak is visible, centred between 800 and 1000 nm (figure 4*e*,*f*). This feature has been attributed to localized surface plasmons,

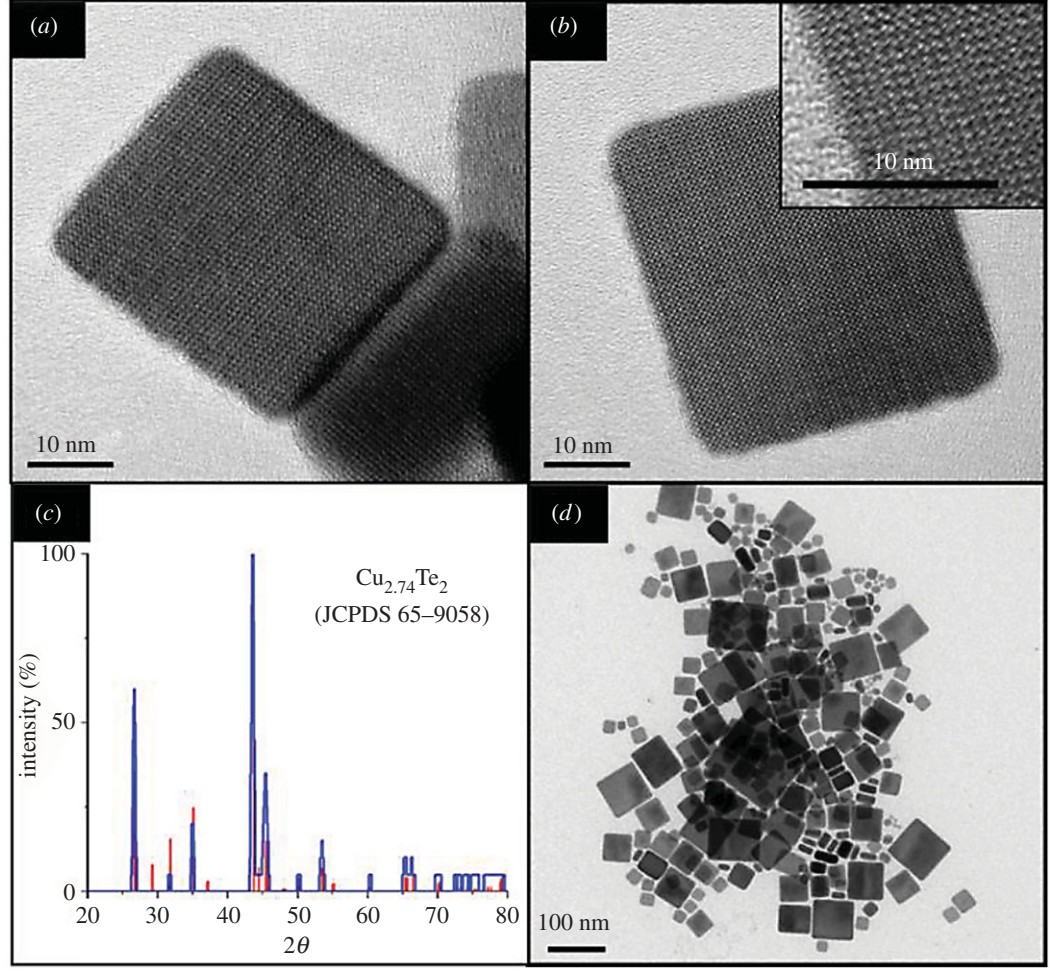

**Figure 3.** Planar squares. (*a,b*) TEMs of the shape controlled rickardite planar squares. Although samples are polydisperse nearly all NCs have the square shape. (*c*) The XRD measurements match well with those of $Cu_{2.74}Te_2$, a tetragonal form of rickardite. (*d*) NCs are up to 50 nm in the length of their principle dimension and several NCs appear stacked on edge with aspect a ratio of up to 4.

which arise due to oxidation of the $Cu^+$ to $Cu^{2+}$ and has been previously found to coincide with the copper-poor stoichiometry of the NCs [27]. The position of the spectral peak has been found to be strongly dependent on particle size [10,27,31] due to the confinement effect on the resonant energy of the surface plasmon in the particle. The band gap is also found to be shifted from the bulk reported values of 0.9–1.2 eV [35]. This is most likely a result of the confinement effect in nanoparticles [36] although slight changes in the crystal structure and stoichiometry can also affect the band structure [35].

Similar nanocrystals, shown in figure 2*b* could also be grown via a one-pot synthesis using this standard reaction in which precursors are added at room temperature. The pot was then heated to reaction temperature, again 180°C. This method produces almost entirely spheroidal nanocrystals but with an average diameter of 6.1 ± 1.1 nm (18%). The improved shape uniformity can be ascribed to greater uniformity of growth conditions after nucleation. The larger distribution in size is typical of many one-pot syntheses, in which nucleation occurs over a longer period of time. Without significant size focusing, larger particles grow from earlier nuclei while later nucleation events spawn smaller particles. The XRD pattern of this material (figure 2*a*) was assigned to that of tetragonal rickardite, $Cu_{2.74}Te_2$, but the decreased crystallinity and resulting missing peaks make this determination difficult.

Very small particles (figure 2*d*) could be grown by performing the standard reaction using pure oleylamine as the solvent, without octadecene or oleic acid present. This reaction produces nanoparticles of approximately 2 nm in size with most particles appearing to be spheroidal in loose clusters or slightly ovular when close packed. Based on XRD measurements (figure 2*c*), these nanocrystals appear to be a highly crystalline orthorhombic form of rickardite, $Cu_{2.86}Te_2$. Furthermore, they appear highly faceted and disk-like in shape, with several apparent 'nanorods' in close-packed formations. This behaviour has been previously observed with oblate spheroids and often indicates

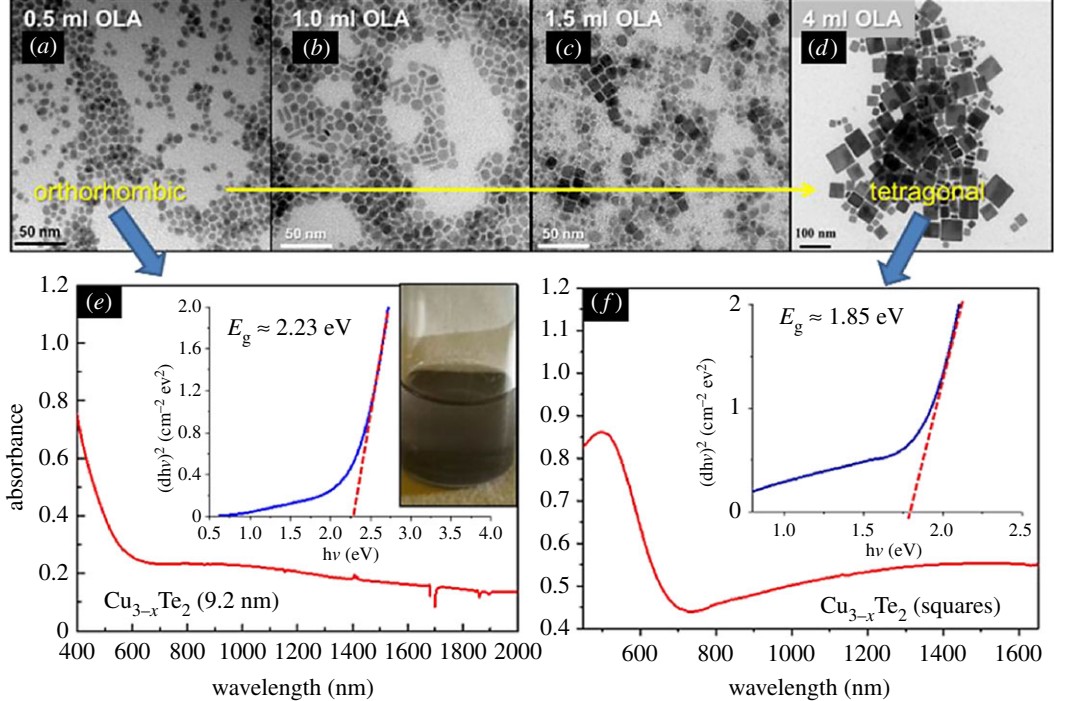

**Figure 4.** Shape control via increased addition of oleylamine. ($a$–$d$): A series of TEMs of $Cu_{3-x}Te_2$ synthesized via the standard reaction with varying volumes of oleylamine added. UV-vis-IR absorption spectra for the standard reaction sample ($e$) and the planar squares ($f$) showing the direct band edge for both materials. The absorbance decays sharply near the band edge at 600–700 nm, which can be fitted more precisely using the linear region of the Tauc plot for both samples (inset). Indirect band gaps were not clearly observed so the near band edge intensity around 1.5 eV is presumed to be due to absorption by optically active traps. The absorbance after 800 nm, peaked at 950 nm (spheroidal) and 1550 nm (squares), is caused by plasmon absorbances. This is due to localized surface plasmons caused by free carriers in the copper telluride, as observed in previous studies [10,31].

**Table 1.** EDX data collected from $Cu_{2.26}Te_2$.

| element | peak area | area sigma | k-factor | weight% | atomic% |
|---|---|---|---|---|---|
| Cu | 10 187 | 185 | 0.007 | 36.02 | 53.07 |
| Te | 12 863 | 245 | 0.010 | 63.98 | 46.93 |

these particles are stacked disks [26,37] with c-axis dimensions of approximately 1 nm, giving a maximum observed aspect ratio of about 2. The change in the size of the particles can be ascribed to the lack of stearic acid. Compared to copper stearate, the copper oleylamine ligand bond should be significantly weaker, permitting the formation of many more nuclei upon injection [38]. This allows the formation of a larger number of nuclei at the injection temperature which in turn restricts the final size of the particles. Furthermore, amines have been observed to effectively etch copper nanocrystals. The size of the nanoparticles is then a steady state function of the competing deposition and etching reactions. In the latter case, the reaction is likely to be highly anisotropic as the amine will preferentially etch copper rich surfaces. The reactivity of the oleylamine with respect to any one facet is then a function of the availability and reactivity of copper at that site (figure 3).

Thus altering the ratio of oleylamine/stearic acid in the standard reaction appears to vastly change the shape of the NCs produced. From figure 4$a$,$b$,$d$, increasing the concentration of oleylamine eightfold in the standard reaction creates a polydisperse sample of planar squares with principle dimensions between 5 and 30 nm. These squares have a crystal structure of tetragonal rickardite, nominally $Cu_{2.74}Te_2$. The stoichiometric ratio of the particles in this sample was observed to be $Cu_{2.26}Te_2$ by EDX measurements, demonstrated in table 1, indicating a large degree of copper oxidation and/or vacancy. The transition from isotropic growth to shape controlled growth can be seen from figure 4$a$ to $d$ where increasing the addition of oleylamine increases directed growth along the [010] and [100] planes.

The formation mechanism for these particles is, as previously assumed, kinetically controlled growth where the oleylamine encourages facet-selective growth along the a-axes. This occurs by etching copper from the copper rich surfaces of the [001] plane and depositing it along the tellurium rich [010] and [100] planes. The absorption profile for the planar squares (figure 4f) is much closer to the bulk values of copper telluride, with a plasmon peak at approximately 1500 nm and an optical band edge around 1.85 eV. Assuming rectangular shapes visible in the sample TEMs to be stacked planar crystals, we observed NCs with aspect ratios of up to 4.

# 4. Conclusion

In conclusion, we have demonstrated a ligand balanced reaction to produce monodisperse samples of $Cu_{3-x}Te_2$, corresponding to tetragonal and orthorhombic forms of rickardite. These can be synthesized with principle dimensions between 1 nm to 30 nm and c/a aspect ratios of up to 4, and are readily dispersible in non-polar solvents. We have also demonstrated the ability to use the oleylamine to stearic acid ratio to control the shape of the NCs, with lower ratios producing small spheroidal faceted NCs and high ratios producing polydisperse planar squares. As these materials are optically active and capable of absorbing wavelengths as low as 1.0 eV, monodisperse samples of rickardite could be used as the absorptive layer in solar cells and photodetectors, for optical switching applications, or as ideal systems for studying surface plasmon effects. Furthermore, square planes of $Cu_{3-x}Te_2$ should display unique optical properties with respect to plasmon absorption in the IR, certainly warranting further investigation. As such, $Cu_{3-x}Te_2$ represents yet another attractive nanocrystalline material for both photovoltaic applications as well as further study of fundamental NC physics.

Data accessibility. This article does not contain any additional data.

Authors' contributions. J.D. and G.J. conceived, designed and conducted the experiments. C.W., P.Y. and R.L. participated in the manuscript preparation. J.L. and W.Z. collected and analysed the data. S.Z. supervised and funded the project. All authors gave final approval for publication.

Competing interests. The authors have no competing interests.

Funding. Financial support for this work was provided by the National Key Research and Development Program of China (no. 2016YFB0301101), the Robert A. Welch Foundation (F-1464), and the National Science Foundation Industry/University Cooperative Research Center on Next Generation Photovoltaics (IIP-1134849).

Acknowledgements. The authors acknowledge Professor Liguo Wang of the Henan Province Industrial Technology Research Institute of Resources and Materials.

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
