## [Reviewer comments · Royal Society Open Science]

Review History

RSOS-181602.R0 (Original submission)

Review form: Reviewer 1 (Yawen Tang)

Is the manuscript scientifically sound in its present form?

Yes

Are the interpretations and conclusions justified by the results?

Yes

Is the language acceptable?

Yes

Is it clear how to access all supporting data?

Yes

Do you have any ethical concerns with this paper?

No

Have you any concerns about statistical analyses in this paper?

No

Recommendation?

Accept with minor revision (please list in comments)

Comments to the Author(s)

Semiconductor nanocrystal (NC) plays important roles in solar cells. In this paper, the authors synthesized monodisperse copper telluride and changed the ratio of oleylamine to stearic acid to study the morphology of final products. Various characterizations were executed to study the structure and composition, and the results could be interest in the readers of Royal Society Open Science. The manuscript can be accepted after a minor revision. Below are my comments:

1. The specific graph order (like a, b...) should be provided in every Figure, to make reading convenient for the readers.
2. In Page 2, Line 48, "The stoichiometric ratio of the particles in this sample was observed to be $\text{Cu}_{2.26}\text{Te}_2$ by EDX measurements", the corresponding EDX pattern needs to be offered.

Review form: Reviewer 2

Is the manuscript scientifically sound in its present form?

No

Are the interpretations and conclusions justified by the results?

No

Is the language acceptable?

Yes

Is it clear how to access all supporting data?

Yes

Do you have any ethical concerns with this paper?

No

Have you any concerns about statistical analyses in this paper?

No

Recommendation?

Reject

Comments to the Author(s)

Dear editor,

The manuscript "Sulfur-Free Synthesis of Size Tunable Rickardite ($\text{Cu}_{3-x}\text{Te}_2$) Spheroids and Planar Squares" has reported a method to synthesize $\text{Cu}_{3-x}\text{Te}_2$ nanoparticles in shapes of Spheroids and Planar Squares. It is claimed tuning the amount of oleylamine as the coordinating ligand is the key to tune the nanoparticle shapes from spheroids to planar squares. The pros of this work is that the TEM images have proved such evaluation process, however, the presented method cannot considered as novel method, since it is slightly different than the methods from previous literature (Inorg. Chem. 2018, 57, 10241–10248; Nature Comm. volume 8, Article

number: 14925 (2017)) and show not much improvement. And the resulted nanoparticles are hardly considered as monodispersed especially for the ones in shape of planar squares (in Figure 4). Adding oleylamine seems to be able to cause the shape evolution, however, loss control of nanoparticle monodispersity and uniformity. Further study to find a better coordinating ligand is recommended.

Decision letter (RSOS-181602.R0)

03-Dec-2018

Dear Dr Du:

Title: Sulfur-Free Synthesis of Size Tunable Rickardite ($\text{Cu}_3\text{-xTe}_2$) Spheroids and Planar Squares
Manuscript ID: RSOS-181602

The editor assigned to your manuscript has now received comments from reviewers. We would like you to revise your paper in accordance with the referee and Subject Editor suggestions which can be found below (not including confidential reports to the Editor). Please note this decision does not guarantee eventual acceptance.

Please submit your revised paper before 26-Dec-2018. Please note that the revision deadline will expire at 00.00am on this date. If we do not hear from you within this time then it will be assumed that the paper has been withdrawn. In exceptional circumstances, extensions may be possible if agreed with the Editorial Office in advance. We do not allow multiple rounds of revision so we urge you to make every effort to fully address all of the comments at this stage. If deemed necessary by the Editors, your manuscript will be sent back to one or more of the original reviewers for assessment. If the original reviewers are not available we may invite new reviewers.

On behalf of the Subject Editor Professor Anthony Stace and the Associate Editor Professor Claire Carmalt.

RSC Associate Editor:
Comments to the Author:
(There are no comments.)

RSC Subject Editor:
Comments to the Author:
(There are no comments.)

Reviewers' Comments to Author:
Reviewer: 1

Comments to the Author(s)

Semiconductor nanocrystal (NC) plays important roles in solar cells. In this paper, the authors synthesized monodisperse copper telluride and changed the ratio of oleylamine to stearic acid to study the morphology of final products. Various characterizations were executed to study the structure and composition, and the results could be interest in the readers of Royal Society Open Science. The manuscript can be accepted after a minor revision. Below are my comments:

1. The specific graph order (like a, b...) should be provided in every Figure, to make reading convenient for the readers.
2. In Page 2, Line 48, "The stoichiometric ratio of the particles in this sample was observed to be $\text{Cu}_{2.26}\text{Te}_2$ by EDX measurements", the corresponding EDX pattern needs to be offered.

Reviewer: 2

Comments to the Author(s)
Dear editor,

The manuscript "Sulfur-Free Synthesis of Size Tunable Rickardite ($\text{Cu}_{3-x}\text{Te}_2$) Spheroids and Planar Squares" has reported a method to synthesize $\text{Cu}_{3-x}\text{Te}_2$ nanoparticles in shapes of Spheroids and Planar Squares. It is claimed tuning the amount of oleylamine as the coordinating ligand is the key to tune the nanoparticle shapes from spheroids to planar squares. The pros of this work is that the TEM images have proved such evaluation process, however, the presented method cannot considered as novel method, since it is slightly different than the methods from previous literature (Inorg. Chem. 2018, 57, 10241–10248; Nature Comm. volume 8, Article number: 14925 (2017)) and show not much improvement. And the resulted nanoparticles are hardly considered as monodispersed especially for the ones in shape of planar squares (in Figure 4). Adding oleylamine seems to be able to cause the shape evolution, however, loss control of

nanoparticle monodispersity and uniformity. Further study to find a better coordinating ligand is recommended.

Author's Response to Decision Letter for (RSOS-181602.R0)

See Appendix A.

Decision letter (RSOS-181602.R1)

14-Jan-2019

Dear Dr Du:

Title: Sulfur-Free Synthesis of Size Tunable Rickardite ($\text{Cu}_{3-x}\text{Te}_2$) Spheroids and Planar Squares
Manuscript ID: RSOS-181602.R1

It is a pleasure to accept your manuscript in its current form for publication in Royal Society Open Science. The chemistry content of Royal Society Open Science is published in collaboration with the Royal Society of Chemistry.

On behalf of the Subject Editor Professor Anthony Stace and the Associate Editor Professor Claire Carmalt.

RSC Associate Editor
Comments to the Author:
(There are no comments.)

Reviewer(s)' Comments to Author:

Appendix A

Response to Referees

Thank you very much for your professional comments. All advices are very important to our studies. These suggestions have great guiding significance for our academic writing and scientific research. We have modified the manuscript accordingly, and the detailed corrections are listed below point by point for the reviewers' comments as following:

Reviewer: 1

Semiconductor nanocrystal (NC) plays important roles in solar cells. In this paper, the authors synthesized monodisperse copper telluride and changed the ratio of oleylamine to stearic acid to study the morphology of final products. Various characterizations were executed to study the structure and composition, and the results could be interest in the readers of Royal Society Open Science. The manuscript can be accepted after a minor revision. Below are my comments:

1. The specific graph order (like a, b...) should be provided in every Figure, to make reading convenient for the readers.

Response:

The graph order has been provided and revised in every Figure in the manuscript.

2. In Page 2, Line 48, "The stoichiometric ratio of the particles in this sample was observed to be $\text{Cu}_{2.26}\text{Te}_2$ by EDX measurements", the corresponding EDX pattern needs to be offered.

Response:

The stoichiometric ratio of the particles in this sample is added and demonstrated in Table 1.

Table 1. EDX data collected from $\text{Cu}_{2.26}\text{Te}_2$

Element	Peak Area	Area Sigma	k-factor	Weight%	Atomic%
Cu	10187	185	0.007	36.02	53.07
Te	12863	245	0.010	63.98	46.93

Reviewer: 2

The manuscript “Sulfur-Free Synthesis of Size Tunable Rickardite ($\text{Cu}_{3-x}\text{Te}_2$) Spheroids and Planar Squares” has reported a method to synthesize $\text{Cu}_{3-x}\text{Te}_2$ nanoparticles in shapes of Spheroids and Planar Squares. It is claimed tuning the amount of oleylamine as the coordinating ligand is the key to tune the nanoparticle shapes from spheroids to planar squares. The pros of this work is that the TEM images have proved such evaluation process, however, the presented method cannot be considered as a novel method, since it is slightly different than the methods from previous literature (Inorg. Chem. 2018, 57, 10241–10248; Nature Comm. volume 8, Article number: 14925 (2017)) and show not much improvement. And the resulted nanoparticles are hardly considered as monodispersed especially for the ones in shape of planar squares (in Figure 4). Adding oleylamine seems to be able to cause the shape evolution, however, loss control of nanoparticle monodispersity and uniformity. Further study to find a better coordinating ligand is recommended.

Response:

Thanks for your valuable comments. In this manuscript, the oleylamine is employed to control the morphology of $\text{Cu}_{3-x}\text{Te}_2$ nanoparticles. We have carefully searched and read the relevant literature and cited (1) Enrico M, Mauro G, Tu RY, Jeremy D, Giovanni B, Roberto G, Luca T, Liberato M. 2018 Ab Initio Structure Determination of Cu_{2-x}Te Plasmonic Nanocrystals by Precession-Assisted Electron Diffraction Tomography and HAADF-STEM Imaging. *Inorg. Chem.* 57, 10241–10248 (doi: 10.1021/acs.inorgchem.8b01445) and (2). Lim J, Schleife A, Smith M 2017 Optical determination of crystal phase in semiconductor nanocrystals. *Nat Commun.* 8, 14849 (doi: 10.1038/ncomms14849).

Inspired by your valuable the suggestion, and we plan to run large number of experiments by using different coordinating ligand on this reaction. Thank you again for this valuable suggestion which have greatly inspired us and brought us new ideas and new research directions.